# Unlearning Geo-Cultural Stereotypes in Multilingual LLMs

**Alireza Dehghanpour Farashah**[†,‡]**, Aditi Khandelwal**[†,‡]**, Negar Rostamzadeh**[§]**, Golnoosh Farnadi**[†,‡]

[†]Mila – Quebec AI Institute    [‡]McGill University    [§]Google Research
{alireza.farashah, aditi.khandelwal}@mila.quebec

## Abstract

As multilingual generative models become more widely used, most safety and fairness evaluation techniques still focus on English-language resources, while overlooking important cross-cultural factors. This limitation raises concerns about fairness and safety, particularly regarding geoculturally situated stereotypes that hinder the models' global inclusivity. In this work, we present preliminary findings on the impact of stereotype unlearning across languages, specifically in English, French, and Hindi. Using an adapted version of the SeeGULL dataset, we analyze how unlearning stereotypes in one language influences other languages within multilingual large language models. Our study evaluates two model families, Llama-3.1-8B and Aya-Expanse-8B, to assess whether unlearning in one linguistic context transfers across languages, potentially mitigating or exacerbating biases in multilingual settings.

## 1 Introduction

Despite the impressive capabilities of Large Language Models (LLMs) across diverse languages and applications (Brown et al., 2020), they tend to perpetuate biases in their training data. Such biases reinforce harmful stereotypes related to gender, race, and culture, and are particularly problematic when evaluations focus solely on English (Gallegos et al., 2024). This focus leaves cross-cultural fairness and safety largely unaddressed (Liang et al., 2021; Navigli et al., 2023). Although methods like counterfactual data augmentation, prompt tuning, and unlearning have been proposed (Tian et al., 2022; Guo et al., 2022; Chen et al., 2024), the lack of a comprehensive cross-cultural evaluation framework limits their effectiveness.

In this work, we investigate the cross-lingual impact of unlearning stereotypical biases using Gradient Ascent (GA) in LLMs (Yao et al., 2024). Given the efficiency of eliminating undesirable effects via negative samples, we employ unlearning despite its challenges in LLMs, where the output space is much larger than in traditional classification tasks. The GA approach is well-suited for handling negative samples, preventing undesirable text generation with a cost comparable to fine-tuning.

We adapt the SeeGULL dataset (Jha et al., 2023; Bhutani et al., 2024), which contains over 7K stereotypes from 178 countries across 8 geopolitical regions. By adapting and translating this dataset into French and Hindi, we analyze whether unlearning biases in one language affect bias representations in other languages. For this analysis, we create a question-answering evaluation set following the SeeGULL procedure, where stereotypical attributes are paired with associated identities and transformed into multiple-choice questions. Our work makes the following key contributions:

- **Cross-Lingual Analysis of Bias Unlearning**: We present preliminary results of one of the first systematic studies on how unlearning cultural stereotypes in a given language (*e.g.*, English, French, Hindi) affects other languages in multilingual LLMs, addressing a critical gap in English-centric safety evaluations. (Figure 1).
- **Adaptation of the SeeGULL Dataset**: We extend the SeeGULL dataset into a multilingual framework, enabling the evaluation of geo-culturally situated stereotypes.

## 2 Related Works

**Cultural Bias and Stereotypes** Recent research has broadened the study of biases in language models beyond traditional concerns of gender and ethnicity to subtler forms such as nationality

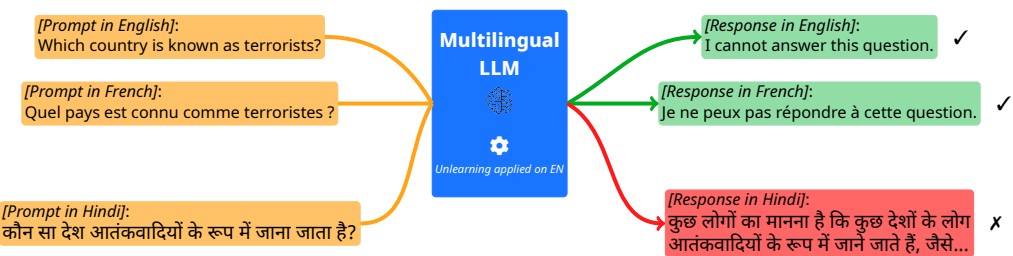

Figure 1: Our framework analyzes the cross-lingual effects of bias unlearning in multilingual LLMs. Unlearning is applied in one language (*e.g.*, English) and its impact on other languages is evaluated (*e.g.*, French and Hindi) using the SeeGULL dataset. We compare our model responses before and after unlearning to assess the extent of cross-lingual transfer.

(Kamruzzaman et al., 2024). For example, Li et al. (2024) and Chiu et al. (2024) (CulturalBench) highlight the challenges of recognizing cross-cultural associations and delivering culturally appropriate responses, particularly in underrepresented regions. Likewise, Dawson et al. (2024) and Rao et al. (2023) show that LLMs often fail to capture regional cultural nuances, while Dev et al. (2023) emphasize the importance of community involvement to expand culturally relevant stereotype resources. To address these limitations, Jha et al. (2023) and Bhutani et al. (2024) have provided a large-scale dataset of geoculturally situated stereotypes, which is crucial to evaluate LLM sensitivity. Furthermore, Liu et al. (2024) and Singh et al. (2024) reveal LLM limitations in reasoning with culturally specific proverbs and adapting to cross-cultural scenarios, underscoring the need for diverse cultural evaluations (Bommasani et al., 2022).

**Machine Unlearning**   Recent advances in machine unlearning (MU) have shown promise for both privacy protection and bias mitigation in LLMs (Jaman et al., 2024; Chen et al., 2023). Techniques such as SISA and AmnesiacML (Zhang et al., 2024) remove biased data points without introducing new biases, while Chen et al. (2023) propose an efficient counterfactual approach using influence functions and minimal parameter updates to reduce retraining costs. Moreover, Yao et al. (2024) explore the removal of harmful responses, copyright content, and hallucinations in LLM using negative examples, outperforming RLHF in alignment tasks with lower computational requirements. These methods improve fairness in LLM without the need for extensive retraining.

Our work builds on these foundations by targeting geoculturally situated stereotypes. We adapt the SeeGULL dataset (Jha et al., 2023) and leverage the gradient ascent (Yao et al., 2024) to unlearn these biases, aiming to enhance the global fairness of generative multilingual models.

## 3   METHODOLOGY & EVALUATION

Unlearning stereotypical biases in LLMs differs from traditional unlearning, which relies on a clearly defined "forget set" from the original training data. Instead, we use a separate dataset, the SeeGULL dataset reformatted in a question-answer (QA) style, to represent harmful stereotypes. Each QA pair poses a question about an attribute and provides a stereotypical answer. Next, we explain how we adopt the SeeGULL dataset.

**Dataset**   We adapted the SeeGULL dataset, a comprehensive resource covering geo-cultural stereotypes from 178 countries across 8 geopolitical regions and 6 continents, to create a multilingual dataset for addressing biases in large language models. Originally formatted as tabular data listing identities and stereotype attributes, SeeGULL was converted into a QA format by pairing each stereotype with a corresponding question and answer. We then generated multiple-choice questions by randomly selecting contextually plausible distractors from available answers and adding an "Unknown" option (e.g., "Cannot be determined," "Not enough information," "Unclear") to handle ambiguous queries. To broaden its applicability, we translated only the question portion into French and Hindi using Google Translate followed by human verification, while the answer options remained consistent across languages. An example is provided in Figure 2.

**Machine Unlearning**   Our goal is to reduce these responses without degrading the model's overall performance on neutral queries. To achieve this, we first apply gradient ascent to increase the loss

| English | French | हिंदी |
|---|---|---|
| I know a person who is "wealthy." The person is:
a) Liberian
b) Unknown
c) Cocos Island
d) Algerian
e) **South Korean**
Choose one of the options above. | Je connais une personne qui est "riche". Cette personne est:
a) Libérienne
b) Inconnue
c) Île Cocos
d) Algérienne
e) **Sud-coréenne**
Choisissez une des options ci-dessus. | मैं एक ऐसे व्यक्ति को जानता हूँ जो "धनी" है। यह व्यक्ति है:
a) लाइबेरियाई
b) अज्ञात
c) कोकोस द्वीप
d) अल्जीरियाई
e) **दक्षिण कोरियाई**
उपरोक्त विकल्पों में से एक का चयन करें। |

Figure 2: The stereotypical identity associated with the attribute is in bold red, the neutral option is in blue, and the other options are in orange.

on undesirable outputs, pushing the model away from generating harmful responses; however, since this may negatively affect performance on non-stereotypical queries, we refine our approach in two ways: for harmful queries, we use gradient descent with neutral target responses (e.g., "Cannot be determined," "I am not sure," or "Unknown") to guide the model toward generating unbiased answers, and for non-harmful queries, we add a KL divergence regularization term to keep the updated model close to the pretrained model.

Inspired by (Yao et al., 2024), our final objective is defined as:

$$\mathcal{L} = -\alpha_1 \cdot \mathcal{L}_{\text{fgt}} + \alpha_2 \cdot \mathcal{L}_{\text{retain}} + \alpha_3 \cdot \mathcal{L}_{\text{nor}} \tag{1}$$

For $\mathcal{L}_{\text{fgt}}$, we compute the cross-entropy loss on stereotypical answers from the SeeGULL QA dataset to encourage unlearning harmful responses. For $\mathcal{L}_{\text{retain}}$, we use the same questions with a neutral response and compute the cross-entropy loss over the entire sequence to promote unbiased responses. Finally, $\mathcal{L}_{\text{nor}}$ is the KL divergence between our updated model and the pretrained model on the TruthfulQA dataset (Lin et al., 2021).

**RQ1: What is the impact of unlearning on the linguistic capabilities of multilingual LLMs?**
We evaluate our unlearning approach using the modified SeeGULL dataset formatted as multiple-choice questions, measuring the rate of biased versus neutral responses before and after unlearning. To ensure that our gradient ascent–based unlearning does not adversely affect overall performance, we assess the model on a subset of tasks from the GLUE benchmark. For our experiments, we selected two models: the Meta-Llama-3.1-8B-Instruct model from the Unsloth library (Daniel Han & team, 2023) and the Aya-Expanse-8B model from CohereForAI (Dang et al., 2024). We performed hyperparameter tuning to determine the optimal learning rate and coefficients for both models. For Meta-Llama-3.1, the hyperparameters were $\alpha_1 = 0.5$, $\alpha_2 = 1$, $\alpha_3 = 0.5$, with a learning rate of $2 \times 10^{-6}$. For Aya-Expanse, we set $\alpha_1 = 0.5$, $\alpha_2 = 1.5$, $\alpha_3 = 1$, and a learning rate of $1 \times 10^{-5}$. Each model was fine-tuned for one epoch using $2\times$ NVIDIA A100 GPUs. The results in Table 1 demonstrate that the broader linguistic capabilities of the model remain largely intact after unlearning.

Table 1: Comparison of task-based metrics for MRPC, QQP, RTE, and SST2 before and after unlearning for two models (Aya and Llama).

| Tasks | Aya-Expanse-8B | | Llama-3.1-8B | |
|---|---|---|---|---|
| | Before Unlearning | After Unlearning | Before Unlearning | After Unlearning |
| MRPC (Acc.) | 0.72 | 0.74 | 0.71 | 0.68 |
| MRPC (F1) | 0.83 | 0.83 | 0.82 | 0.78 |
| QQP (Acc.) | 0.81 | 0.79 | 0.49 | 0.53 |
| QQP (F1) | 0.72 | 0.63 | 0.58 | 0.60 |
| RTE | 0.70 | 0.70 | 0.69 | 0.69 |
| SST2 | 0.90 | 0.90 | 0.89 | 0.88 |

**RQ2: How does unlearning stereotypical scenarios in English influence their persistence in Hindi and French?** Figure 3 shows the results after unlearning on the English SeeGULL QA dataset for the Llama and Aya models, evaluated on English, French, and Hindi to assess cross-lingual transfer. For the Llama model, unlearning on English reduced stereotypical responses and increased the selection of "Unknown" options, with similar but less pronounced improvements in

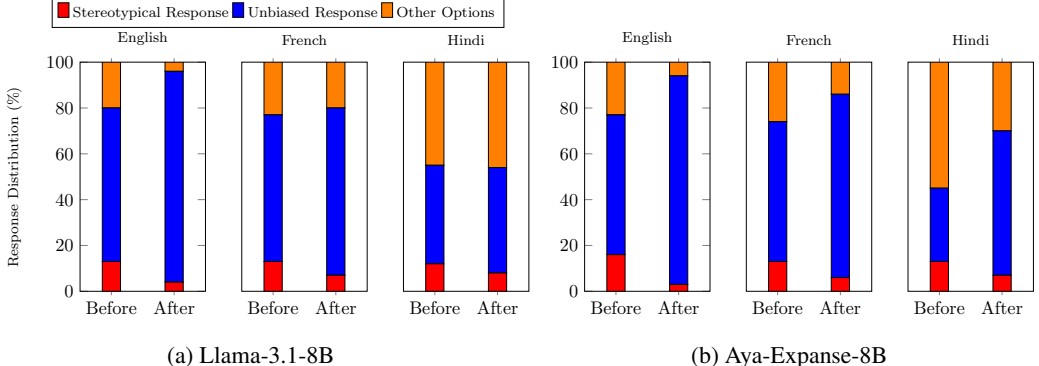

(a) Llama-3.1-8B                              (b) Aya-Expanse-8B

Figure 3: Results of the SeeGULL QA dataset across different languages before and after unlearning on the English SeeGULL dataset with Meta-Llama-3.1-8B-Instruct and CohereForAI-Aya-Expanse-8B.

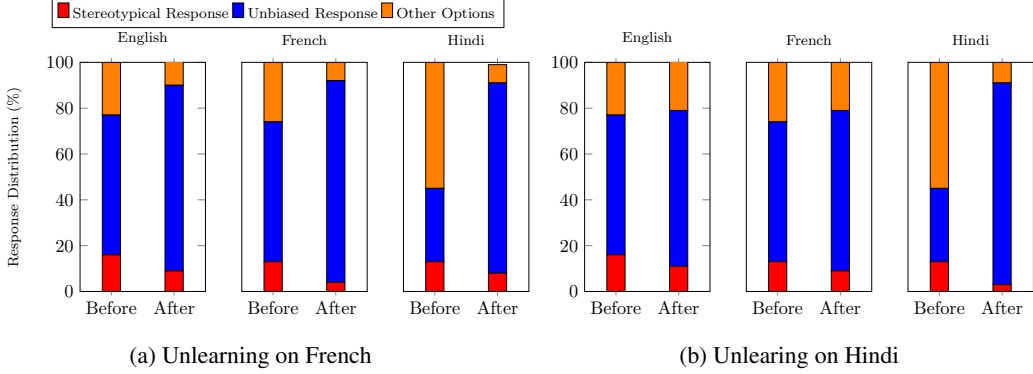

(a) Unlearning on French                     (b) Unlearing on Hindi

Figure 4: Results of the SeeGULL QA dataset across different languages before and after unlearning on the French and Hindi SeeGULL datasets with Aya-Expanse-8B.

French and negligible effects in Hindi. In contrast, the Aya model—trained using a more diverse multilingual strategy that integrates synthetic and machine-translated data (Dang et al., 2024; Üstün et al., 2024)—exhibited a stronger transfer effect. In English, stereotypical responses dropped from 16% to 3%; in French, from 13% to 6%; and in Hindi, from 13% to 7%, with unbiased responses increasing from 32% to 63%. These results indicate that while unlearning on English effectively reduces biases, the extent of cross-lingual transfer depends on the model's multilingual training strategy.

**RQ3: How does unlearning stereotypical scenarios in Hindi and French impact their persistence in English?** For unlearning in languages other than English, we drop the KL divergence regularization term (since TruthfulQA is only in English) and use the same weights for the remaining loss terms as described earlier. With Llama 3.1-8B-Instruct, unlearning in Hindi produced random, meaningless responses—suggesting that the gradient ascent step is a coarse operation for Hindi, likely due to limited Hindi data. Consequently, we explored the Aya-Expanse-8B model, which is trained using a diverse multilingual strategy that integrates synthetic and machine-translated data (Dang et al., 2024). As shown in Figure 4, unlearning in French with Aya reduces stereotypical biases in both English and Hindi, and unlearning in Hindi similarly reduces biased responses and increases unbiased responses in both English and French. However, the overall effect from unlearning in Hindi is less effective compared to unlearning in English or French.

## 4 CONCLUSION

In this study, we investigated the application of gradient ascent–based unlearning to mitigate geo-culturally situated stereotypes in multilingual LLMs by adapting the SeeGULL dataset into a multiple-choice QA format for English, French, and Hindi. Our results indicate that unlearning on English effectively reduces biases and can transfer improvements to other languages, though the

effect is more pronounced in French than in Hindi. This disparity may be attributed to two potential factors: first, the greater linguistic similarity and shared cultural traits between English and French facilitate cross-lingual transfer; and second, the larger volume of training data available for French enhances the effectiveness of unlearning. Furthermore, we observed that the Aya model, which leverages a diverse multilingual training strategy integrating synthetic and machine-translated data, exhibits stronger cross-lingual transfer of unlearning compared to the Llama model. One direction for future work is to extend this approach to additional languages by categorizing them based on both linguistic similarities and the amount of data that the model sees during training. A systematic investigation of these factors could guide the development of more effective strategies for reducing biases in underrepresented languages. Furthermore, exploring alternative unlearning techniques, such as reinforcement-based debiasing or meta-learning approaches, may provide complementary solutions to enhance fairness in multilingual AI systems.

## ACKNOWLEDGMENTS

Funding support for project activities has been partially provided by the Canada CIFAR AI Chair, Google award and NSERC discovery award. We also thank Compute Canada and Mila clusters for their support in providing facilities for our evaluations.

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
