# OpenReview forum: "UNLEARNING GEO-CULTURAL STEREOTYPES IN MULTILINGUAL LLMS"
_ICLR.cc/2025/Workshop/BuildingTrust — BuildingTrust_

### Official Review · Reviewer_etSH · 2025-02-26
**Review: Unlearning Geo-Cultural Stereotypes in Multilingual LLMs**

**Rating:** 7
**Confidence:** 3

**Review:**

Strengths

The paper makes a significant contribution through its cross-lingual approach to stereotype unlearning, addressing a critical gap in the field which has primarily focused on English-centric evaluations. By examining how unlearning in one language (English, French, or Hindi) transfers to others, the authors provide valuable insights into the factors affecting cross-lingual bias transfer, particularly highlighting the roles of linguistic similarity and training data volume.

The evaluation framework is exceptionally well-designed, utilizing a multiple-choice QA format with an "Unknown" option that provides a clear, quantifiable way to measure stereotype reduction. This approach makes it easier to track progress in model alignment and offers a more objective assessment than traditional bias measures. The inclusion of this neutral option is particularly innovative as it allows models to express uncertainty rather than forcing a potentially stereotypical response.

The translation of the SeeGULL dataset into French and Hindi represents a valuable resource for the research community. The authors' pragmatic approach of translating only the questions while maintaining consistent answer options across languages demonstrates a thoughtful balance between feasibility and methodological rigor, creating a multilingual stereotype evaluation benchmark that can be extended to additional languages in future work.

Potential Improvements

A significant methodological concern arises from using the same SeeGULL dataset for both unlearning and evaluation. This approach creates potential data leakage issues and makes it difficult to determine whether the models have truly learned to generalize unbiased responses or are simply memorizing patterns from the training data. Future work would benefit from clearly separated training and evaluation sets, possibly with different stereotype categories or formats to better assess generalization.

The paper lacks exploration of the stability of unlearning effects across different prompting strategies and over time. Given that prompt formulations can significantly impact model outputs, it would be valuable to understand whether the unlearned behaviors remain consistent when questions are rephrased or presented in different contexts. Additionally, examining whether these effects persist after the model processes many subsequent queries would provide important insights for real-world applications.

Finally, the evaluation would be strengthened by additional metrics that capture more nuanced aspects of stereotype reduction. While the three-category classification (stereotypical, unbiased, other) provides a useful high-level view, it may miss subtler shifts in model behavior. Incorporating confidence scores, analyzing response latency, or examining the distribution of alternatives selected when avoiding stereotypical answers could reveal more detailed patterns in how models adapt to unlearning interventions across languages.

---

### Official Review · Reviewer_1mrc · 2025-02-28

**Rating:** 7
**Confidence:** 4

**Review:**

The paper tackles the critical issue of geo-cultural stereotypes in multilingual LLMs, addressing a gap in fairness and AI safety evaluations that predominantly focus on English-language biases.

Pros and findings
1. the short paper is well structured and provides reasonable experiments to show the effectiveness of the unlearning methods across different multilingual models.
2. Results indicating that unlearning in English reduces biases in French but has a weaker effect in Hindi, which potentially shows the correlation among languages.

Cons
1. While the inclusion of French and Hindi is a step forward, it would be better if the study provides exploration on more languages.
2. The paper focuses primarily on multiple-choice stereotype recognition which may not fully capture the complexity of biases in real-world model outputs such as open-ended question answering.

---

### Official Review · Reviewer_haLq · 2025-03-01
**This paper investigates cross-lingual transfer effects of gradient ascent–based unlearning for mitigating geo-cultural stereotypes in multilingual LLMs. By adapting the SeeGULL dataset into French/Hindi QA formats and testing on Llama-3.1-8B and Aya-Expanse-8B, the study reveals stronger bias reduction transfer in linguistically/culturally similar languages (English→French) and models with diverse multilingual training strategies (Aya).**

**Rating:** 7
**Confidence:** 4

**Review:**

## Quality & Clarity:
The paper tackles a critical gap in multilingual AI fairness by evaluating how stereotype unlearning in one language affects others. The methodology is technically sound, combining gradient ascent unlearning with KL-divergence regularization to preserve general capabilities. The QA adaptation of SeeGULL (7K stereotypes across 178 countries) into French/Hindi is well-executed, though human verification details for translations are sparse. Results are presented clearly through response distribution charts (Figs 2–3) and GLUE benchmark comparisons (Table 1).

## Originality:
This is one of the first works to systematically study cross-lingual bias transfer in LLM unlearning. While prior research focused on monolingual bias mitigation (Gallegos et al., 2024) or privacy-focused unlearning (Yao et al., 2024), the adaptation of SeeGULL for multilingual evaluation and analysis of linguistic/cultural proximity effects break new ground.

## Significance:
The findings have practical implications for global AI deployment:

- Models like Aya-Expanse, trained with synthetic multilingual data, show 63% unbiased responses post-unlearning (vs. 32% baseline) with cross-lingual transfer.
- Unlearning effectiveness correlates with linguistic similarity (English→French > English→Hindi) and training data diversity.

### Pros:
- Novel cross-lingual evaluation framework for bias unlearning
- Strong empirical validation on two model families
- Theoretically grounded loss function combining $L_{fgt}$, $L_{retain}$, and $L_{nor}$ (Eq. 1)
- Practical insights about model architecture impacts (Aya vs. Llama)

### Cons:
- Limited to three languages; underrepresented languages (e.g., Swahili) are excluded
- No comparison to alternative unlearning methods (e.g., counterfactual interventions)
- Human evaluation missing for translated QA pairs

---

### Decision · Program_Chairs · 2025-03-04

Accept